# Deep, complex, invertible networks for inversion of transmission effects in multimode optical fibres

**Oisín Moran,[1] Piergiorgio Caramazza,[2] Daniele Faccio,[2] Roderick Murray-Smith[1,*]**
[1]School of Computing Science, University of Glasgow, Scotland.
`oisin@inscribe.ai, Roderick.Murray-Smith@glasgow.ac.uk`,
[2]School of Physics and Astronomy, University of Glasgow, Scotland.
`piergiorgio.caramazza@gmail.com, Daniele.Faccio@glasgow.ac.uk`

## Abstract

We use complex-weighted, deep networks to invert the effects of multimode optical fibre distortion of a coherent input image. We generated experimental data based on collections of optical fibre responses to greyscale input images generated with coherent light, by measuring only image amplitude (not amplitude and phase as is typical) at the output of $1\,\mathrm{m}$ and $10\,\mathrm{m}$ long, $105\,\mathrm{\mu m}$ diameter multimode fibre. This data is made available as the *Optical fibre inverse problem* Benchmark collection. The experimental data is used to train complex-weighted models with a range of regularisation approaches. A *unitary regularisation* approach for complex-weighted networks is proposed which performs well in robustly inverting the fibre transmission matrix, which is compatible with the physical theory. A benefit of the unitary constraint is that it allows us to learn a forward unitary model and analytically invert it to solve the inverse problem. We demonstrate this approach, and outline how it has the potential to improve performance by incorporating knowledge of the phase shift induced by the spatial light modulator.

## 1 Introduction

The ability to better transmit images over multimode fibre (MMF) has applications in medicine, cryptography and communications in general. However, as pointed out by Stasio [2017], MMFs are not normally utilised for imaging because they do not act as a relay optical element. This makes their use for focussing and imaging impractical, without sophisticated compensation for their transmission properties. In this paper we demonstrate that a deep network combining multiple complex dense layers with orthogonal regularisation and conventional autoencoders can successfully invert speckle images generated over a $10\,\mathrm{m}$ long distorted $105\,\mathrm{\mu m}$ MMF without phase information. Previous work has either required phase information, or been limited to short (e.g. $30\,\mathrm{cm}$) and straight fibres.

### 1.1 Deep learning challenges

This particular imaging application has a number of features which are potentially challenging for the machine learning community. The speckle images used as inputs have a non-local relationship with the pixels in the inverted image, making local patch-based approaches impossible, and leading to challenging memory problems. The statistics of speckle images are very different from typical images, meaning that 'off-the-shelf' deep convolutional network approaches—which assume locally-spatial structure—cannot be applied directly. There are clear circular correlations in the images, but finding these requires solving the inversion, leading to a chicken-and-egg problem. The non-locality also means that position-invariant approaches like convolution layers and max-pooling should not be applied before having brought the image back to an appropriate spatial arrangement

From our knowledge of the optics, we know that the transformation of the image to speckles can be represented by a complex-valued, orthogonal transmission matrix $T$, but this matrix will be very large (e.g. for $350 \times 350$ resolution images in and out, $T$ will have $350^4 \approx 15$ billion entries). The longer and more distorted the optical fibre, the smaller the speckles at the output, so the higher the camera resolution required to capture the details.

For instrumentation convenience and cost in real-world applications, we wish to see how far we can go with amplitude-only sensing, i.e. we do not want to have the complexity of using an interferometer, which means we must assume no phase information for the speckle outputs. This means our model fitting of the transmission process is significantly underconstrained.

## 2    Background

### 2.1    Modelling optical fibre transmission

The transmission of optical information by means of a guiding system has always been of paramount interest, e.g. for medical and communication systems. One particular challenge has been the realisation of ever thinner waveguides able to transmit actual images. This could be decisive whenever we want to obtain images of otherwise inaccessible zones, for example due to limitations related to light depth penetration. However, when a wave is confined its propagation is constrained to follow a precise mathematical relationship. In other words, the confining system permits only certain possible wave propagations, called *'modes'*. These constitute an orthogonal basis for the space of all the possible solutions of the propagation.

*Single-mode* fibres can be considered for the transmission of images, but as a single-mode waveguide can carry just a single channel of amplitude information (and a phase one, since light propagation is complex), an image would require either a single-mode fibre bundle or a scanning system [Seibel et al., 2006]. To date, the minimum diameter size for both of these systems is around $1\,\mathrm{mm}$.

*Multimode* fibres (MMF) have clear advantages because, as pointed out in [Choi et al., 2012b], the density of modes for unit of area is 1–2 orders of magnitude greater than that of a fibre bundle. Moreover, whenever using coherent light, single-mode fibres tend to couple between each other if too close (i.e. light crosses over from one fibre to the others), blurring the image and reducing its contrast. The challenge with multi-mode fibres is that since different modes propagate with different velocities in these types of fibre, interference between these generates a scrambled image, as shown in Figures 1 and 3.

In principle, the ability to perfectly model the transmission through an optical fibre would allow imaging through MMFs. The modes of the fibre form an orthogonal basis for the fibre, which can be considered a linear system. If for each input (mode), the output can be calculated, the transfer function of this linear system can be constructed, allowing us to infer an unknown input for a given system output. However, in real optical fibres, mechanical deformations of the fibre, surface roughness, bending and even extremely small temperature variations lead to mode-to-mode coupling and changes in the refractive index, which makes it very challenging to predict the output field analytically or numerically. Nonetheless, however complex, the actual propagation still remains linear and deterministic. Therefore, similarly to complex random media, it has been demonstrated that it is possible to build up an input orthogonal basis such that, by acquiring the output amplitude and phase relative to each mode, we can in principle, identify the *transmission matrix* (TM), $T$, that maps the input into the output [Čižmár and Dholakia, 2011, 2012].[1]

Other techniques have achieved scanner-free endoscopy reconstruction (on 1m long fibre). Choi et al. [2012a] present an empirical approach but based on measurement of both amplitude and phase of the output light field, and requiring 500 measurement repetitions at different incident angles, and improved the resolution [Mahalati et al., 2013] with just single-pixel measurements relative to a sequence of random illuminations, and implemented phase conjugation [Papadopoulos et al., 2012] that allows the restoring process to proceed without calculating the full TM. Finally, a step further was the realisation of a procedure that could include possible bends in the fibre. This was realised in

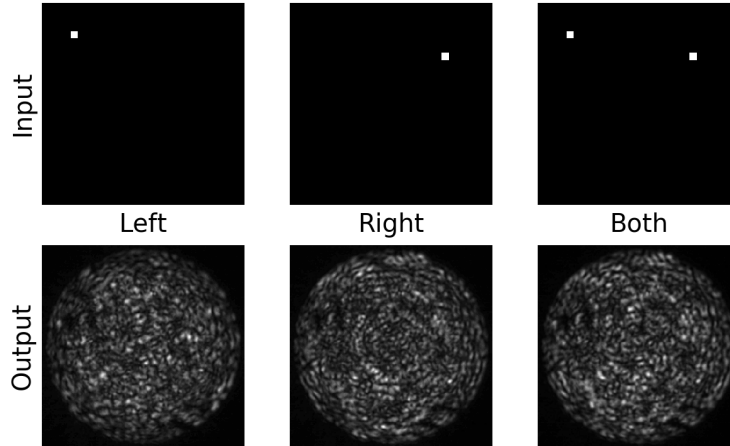

Figure 1: Image of an input of single pixels and their speckle images at the output. Note how a single pixel generates a response over the full space of the output sensor, and at much finer detail than the resolution of the pixel. Also, the amplitude of output of the two-pixel example is not the sum of amplitudes of the single pixel responses, due to complex-valued interactions of modes in the fibre.

[Plöschner et al., 2015] by having a precise characterisation of the fibre and an accurate theoretical model, for straight lengths up to $30\,\mathrm{cm}$.[2]

## 2.2 Related machine learning background

### 2.2.1 Complex deep networks

Hirose [2003, 2012] reviews the long history of the field of complex neural networks. More recent work includes [Tygert et al., 2016], and [Trabelsi et al., 2018], the latter of which provides and rigorously tests the key atomic components for complex-valued deep neural networks—including complex convolutions, complex batch-normalisation, complex weight initialisation, and complex activation functions. Guberman [2016] investigates the difficulties in training complex-valued models due to the lack of order over the complex field and finds them significantly less vulnerable to over-fitting.

## 2.3 Inversion

There are two approaches to using machine learning to solve the inverse problem. We can identify the 'causal' or 'forward' model from image to speckle, then numerically invert the forward model and optimise to find the input most likely to have generated that image. The alternative we test in this paper is whether we can directly learn an inverse model. This approach has been used in the past in a variety of applications, e.g. control [Cabrera and Narendra, 1999], and single-pixel imaging [Higham et al., 2018], and the two approaches can be combined as illustrated in human motor control [Wolpert and Kawato, 1998]. There has been growing interest in invertible neural networks recently [Dinh et al., 2016, Ardizzone et al., 2018, Grathwohl et al., 2018]. In this paper we explore both forward and inverse modelling approaches to complex-valued inversion.

## 3 Experimental setup

The experiment is carried out with the setup illustrated in Figure 2. In order to generate greyscale images, a continuous wave (CW) laser source, with wavelength $\lambda = 532\,\mathrm{nm}$, is used. The laser light

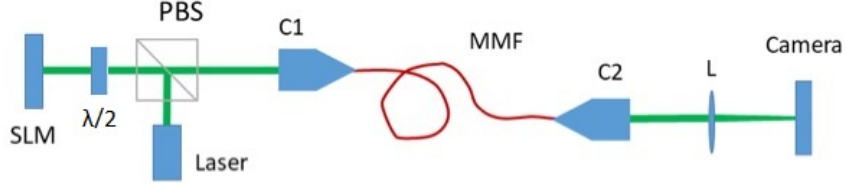

Figure 2: Experimental configuration

beam is modulated in amplitude by means of a phase-only spatial light modulator (SLM) along with a polarised beam-splitter (PBS) and a half-wave plate ($\lambda/2$) in the configuration reported in Figure 2. The SLM is controlled by a computer which generates the hologram that will be projected onto it. In this way, the resulting greyscale image, with values ranging from 0 to 100, is coupled into a multimode step-index fibre by means of a doublet collimator lens (C1). At the output of the fibre, another collimator (C2) along with a second lens (L) are used to image the output of the fibre onto a CMOS camera. The data acquired with the camera are digitised in the range from 0 to 255. Moreover, the pixel resolution of a single acquisition is $350 \times 350$. Our field of view is reduced by the output lenses. However, in principle, it is possible to remove the lenses and expect the neural network to include the free-air propagation operator, from the fibre output to the camera, in its learning process as well. For this experiment two multimode fibre (core diameter = $105\,\mu\text{m}$ and NA = 0.22) with different lengths, respectively $1\,\text{m}$ and $10\,\text{m}$, have been used. The sensor noise has an average error of ca. 1% of total signal per pixel.

The input images have a resolution of $28 \times 28$ pixels. Several different image datasets were used: MNIST [Lecun et al., 1998], Fashion-MNIST [Xiao et al., 2017] and random images. MNIST is the common dataset containing ten classes of handwritten digits (0 to 9), whereas fashion MNIST consists of ten different classes of clothing. As introduced before, all input images values are in the range: 0 to 100. In order to give an example of the input and output images, we can refer to Figure 3. The acquisition procedure is very straightforward: once the first hologram is loaded on the SLM, an image is captured at the output of the fibre. Then, we move to the second hologram and so on for the entire training and testing dataset. Finally, we want to pinpoint that our experiment is not relying on any phase acquisition, so that training and testing process are implemented with just amplitude images. In the same way, no scanning procedure has been applied.

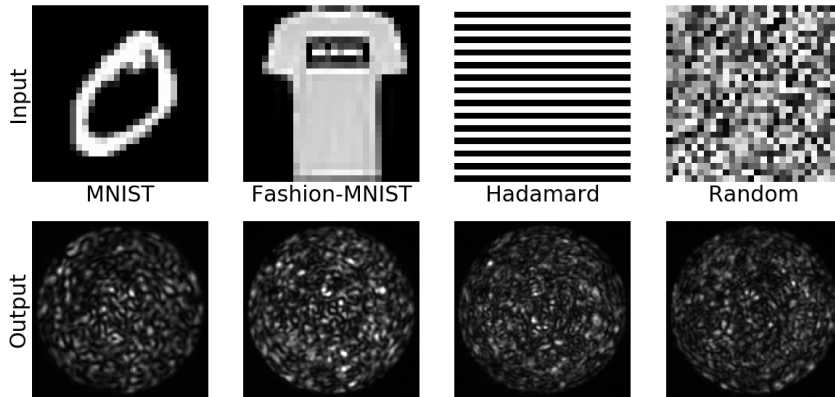

Figure 3: Examples of input and output (speckle) images from the experiment. The inference task is then to regenerate the input image from the speckle image.

## 3.1 Dataset description

The dataset generated with the experimental equipment is composed of several public sets, including: MNIST [Lecun et al., 1998], Fashion-MNIST [Xiao et al., 2017], and some images of Muybridge's historic stop-motion films [Muybridge, 1955]. Examples of input image and speckle response are shown in Figure 3. Further we generated datasets composed of $N_i^2$ binary Hadamard bases, and 60,000 random binary images.

### 3.1.1 *Optical fibre inverse problem* **Benchmark collection**

We share this dataset of of 90,000 images repeated at 4 fibre lengths. The images are acquired at a fibre length of $1\,\mathrm{m}$ and $10\,\mathrm{m}$. Input images are at $28 \times 28$ pixel resolution for compatibility with the widely used MNIST and MNIST-fashion benchmarks. Speckle images are recorded at $350 \times 350$ pixels. We provide accompanying code and sample models, with the intention that this can be used as a benchmark for this type of inference—for both the machine learning and optics communities.

## 4 Models used

Our approach was to transform the speckle image back towards the original image space with a complex layer, as in Figure 4,[3] followed by a denoising autoencoder which can compensate for imperfect reconstruction of the target image. Table 1 outlines the model performances.

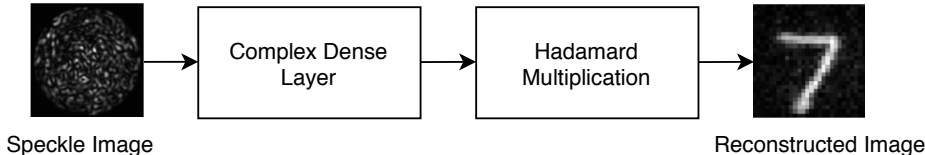

Figure 4: Inversion pipeline. A speckled image is transformed to the inferred input image that generated it, via an initial complex affine transformation.

### 4.1 Nonlinear compensation for SLM effects and laser power drop-off

The spatial light modulator (SLM) in the configuration showed in Figure 2 introduces nonlinear effects as a function of pixel value to intensity and phase (Figure 5). We characterise the intensity using a photodiode (after the PBS). The phase is obtained by splitting the beam reflected by the SLM (after the PBS) in two and making the beams interfere, fixing the pixel value of one of the beams and varying the pixel value of the other one between 0-100, and measuring the result with a camera. The intensity of the laser has a roughly Gaussian decay as we move away from the focal point, requiring another nonlinear layer to learn this effect. We use a general Hadamard elementwise multiplication layer to capture these drop-off effects and any scaling issues needed in the unitary regularisation case.

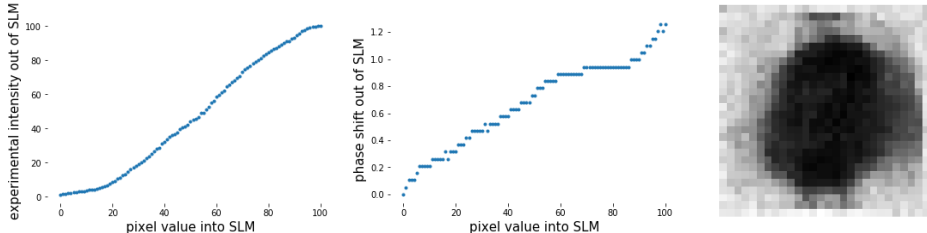

Figure 5: Optical nonlinearities. Graphs of Experimental Intensity and Phase mappings induced by Spatial-light modulator as a function of pixel value. Right, parameters of the Hadamard layer show the learned function of intensity drop-off of the laser

### 4.2 Complex dense layer for inverting $T$

As described earlier, we know the transmission matrix $T$ is complex-valued, so implementation of the initial dense layer of the network with complex-valued weights is appropriate. Trabelsi et al. [2018] investigated the usage of various different complex-valued activation functions, namely:

Complex ReLU or $\mathbb{C}$ReLU,[4] modReLU [Arjovsky et al., 2016], and $z$ReLU [Guberman, 2016] and found that models using the $\mathbb{C}$ReLU function had a lower classification error on all three tested datasets—CIFAR-10, CIFAR-100 and SVHN* (a reduced training set of SVHN). Following this we decided to focus on activation functions with separable real and imaginary parts, focusing in particular on complex-valued analogues of the standard variants of the ReLU, such as the PReLU [He et al., 2015]. This is convenient, as if the complex numbers are represented as two real-valued channels these complex-valued activation functions are easily computable by performing a standard real-valued activation on each channel separately. The specific complex-weighted network used in this paper was implemented with Keras [Chollet et al., 2015] and TensorFlow [Abadi et al., 2015]. This layer tends to be a memory-expensive element, scaling as $O(N_i^2 N_o^2)$ and, due to the non-local nature of the inversion process, we cannot work on rectangular sub-patches of the speckle images.

## 4.3 Weight regularisation approaches

### 4.3.1 Unitary, complex dense layer regularisation

Orthogonal regularisation of weights was introduced in [Brock et al., 2016] to ensure an efficient use of the representational capability of the layer. Our motivation is related to the physics of the problem which suggest that the transmission channel can be represented by a limited number of orthogonal modes. For the fibre used in this experiment there are 9000 modes. We therefore extend Brock et al.'s approach to the complex-valued domain, by proposing a *unitary regularisation*, where the complex weight matrix $W$ of the complex layer is pushed towards a unitary matrix for $W \in \mathbb{C}^{m \times m}$, or more generally toward a semi-unitary matrix for rectangular $W$, by a regularising term

$$\mathcal{L}_{unitary}(W) = \|WW^* - I\|_1 \text{ for } W \in \mathbb{C}^{m \times n}, \tag{1}$$

where $W^*$ is the conjugate transpose of $W$.

### 4.3.2 Amplitude and phase weight regularisation for complex layer

As part of the implementation of the complex-weighted dense layer, we can choose different options for weight regularisation. One such option is amplitude & phase regularisation, implemented as a weighted sum of amplitude and phase penalisation terms with two parameters $\alpha_r$ and $\alpha_\phi$ to trade-off phase and amplitude penalties: $\mathcal{L}(W, \alpha_r, \alpha_\phi) = \sum_i \alpha_r |W_i|^2 + \alpha_\phi \underline{/W_i}$. Amplitude-only regularisation ($\alpha_\phi = 0$) is equivalent to the standard per-channel $l_2$ penalisation applied to both channels independently: $\mathcal{L}(W) = \sum_i (\Re\{W_i\}^2 + \Im\{W_i\}^2) = \sum_i |W_i|^2$. Phase regularisation could be useful for tasks with a target phase, or where something is known about the phase characteristics of the system involved. However, it is not explored in this paper.

## 4.4 Inversion of the forward function

A complex square matrix $W$ is unitary if its conjugate transpose $W^*$ is its inverse, so if we enforce unitarity in the complex dense layer, we automatically have an analytic inverse at negligible computational cost, and we can directly optimise the parameters in both a *forward* and *inverse* manner, as shown in Figure 6. We do this by creating an autoencoder-like structure for the forward model, going from image to speckle by multiplying by our estimate of complex matrix $T$, and then back to the original image again via its inverse $T^*$. This has the advantage of being able to incorporate input phase information into the optimisation of the model parameters for the forward path. As discussed in section 4.1, the SLM induces a phase shift which is a nonlinear function of pixel amplitude, as well as a nonlinear modulation of the pixel amplitude itself. Future physical experiments could use this approach to explicitly manipulate phase information on the input to better identify $T$, without having to measure output phase on the speckles. The inverse path goes from speckle to image, but has no phase information (although there is the option of including the inferred phase from the forward path, shown by a dotted line in Figure 6). This model was implemented in keras/tensorflow, and produced successful inversions when trained simultaneously in forward/backward directions. As our GPU memory was insufficient to allow more than $56 \times 56$ transmission matrices in this more complex model, the results did not improve over the direct inverse method at the higher 112 pixel resolution.

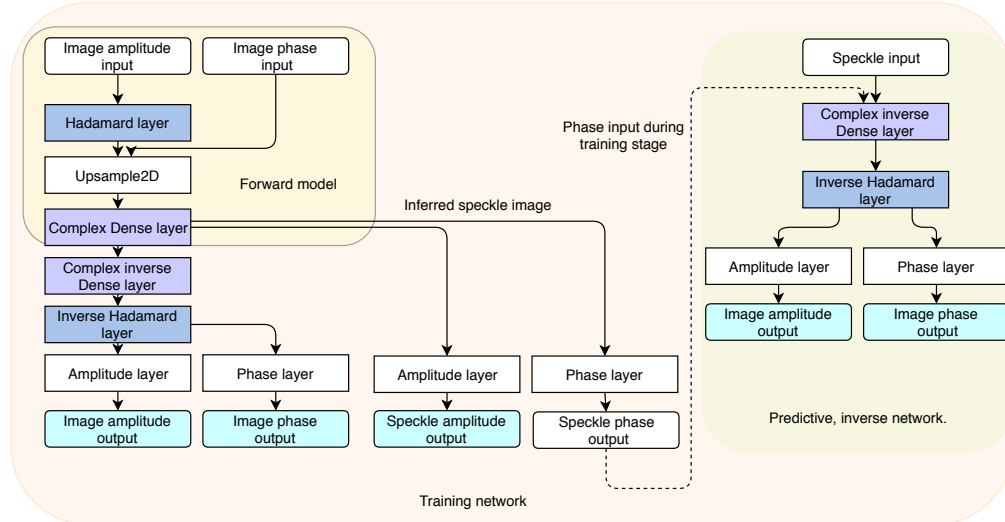

Figure 6: Model structure of the forward and inverse model. Cyan outputs are those for which we have target training data. Coloured layers indicate tied weights between associated forward and inverse layers. During training, inferred speckle phases from the forward model can potentially be used to augment training via the inverse model. Controlling image phase input (much easier than sensing phase output) can also augment learning in this approach.

## 5 Experimental results

In our experimental analysis we first compare approaches to invert the complex transmission matrix, then investigate how we can further refine these images. All results here are on test data not used during training. We split the data into 80% for training, 20% for test.

### 5.1 Inversion of the transmission matrix

Here we compare a real-valued baseline with a complex-valued dense layer with and without $l_2$ weight regularisation and with unitary regularisation. We also used a multiscale approach where multiple average pooling layers after the complex dense layer fed into an output vector composed of a pyramid of halving resolutions of the target images ($28 \times 28, 14 \times 14$ and $7 \times 7$). The results are summarised in Figure 7 and Table 1. The complex-weighted models consistently learn faster and to a better accuracy than real-weighted ones. The evidence for differences among the various regularisation techniques is not strong for this dataset. Although there is some numerical difference in MSE, the visual difference in Figure 7 is minimal.

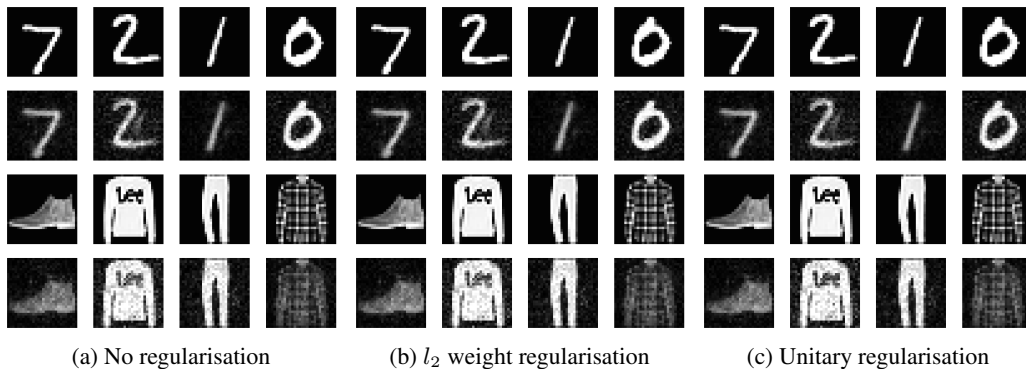

(a) No regularisation      (b) $l_2$ weight regularisation      (c) Unitary regularisation

Figure 7: The impact of regularisation on inference by a single complex-valued layer with various regularisation methods. Speckle images of $112 \times 112$ resolution were used on 1m data.

Table 1: Model comparisons where each single layer model with 19,669,776 parameters (9,835,280 for real-valued models) was trained for 300 epochs, or until convergence, on a speckle resolution of $112 \times 112$, with $\lambda = 0.03$ where the model was $l_2$ regularised.

| Model | | |
|---|---|---|
| Name | Regularisation | MSE |
| Real-valued | $l_2$ weight regularisation | 1034.62 |
| Real-valued | None | 1025.96 |
| Complex | Unitary regularisation | 989.28 |
| Complex | Multiscale no regularisation | 988.10 |
| Complex | $l_2$ weight regularisation | 962.33 |
| Complex | None | 960.31 |

### 5.1.1 Comparison of MSE and SSIM results

While a mean squared error cost function is commonly used, there is often a mismatch between MSE and the perceived quality of an image. Zhao et al. [2015] provides a more in-depth analysis of using perceptually-based loss functions in neural networks. We compared the perceptually-motivated SSIM [Wang et al., 2004] with MSE and found that for poorer quality models it denoised somewhat, but did not add much subjective value to the optimised models for this application.[5]

### 5.1.2 Impact of speckle resolution

We evaluate the impact of varying speckle image resolution for fixed target image resolution $(28 \times 28)$. We test on speckle inputs of 14, 28, 56, 112, & 224 square, shown in Figure 8. Inversion was by a single complex-valued layer minimising MSE, with $l_2$ amplitude weight regularisation. The quality of the final estimate of the inverted image increases steadily with increasing speckle information, even when at 4 or 8 times the target image resolution. This is critical for inversion of longer transmission fibres, as the size of speckles decreases the longer and more distorted the optical fibre.

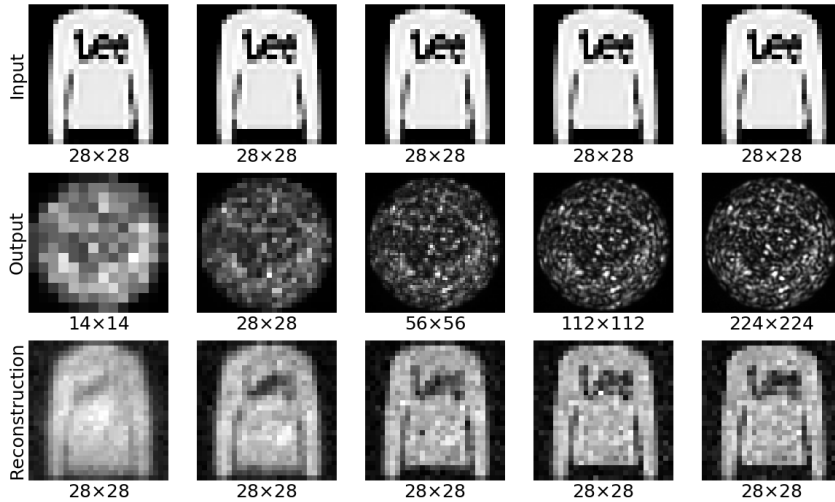

Figure 8: Impact of speckle resolution on quality of inferred image. From left to right we test on speckle image inputs of $N_i =$14, 28, 56, 112, and 224 pixels. This highlights the challenge to the machine learning community—increasing resolution far beyond the target resolution will improve the accuracy of the inverted images (up to the capacity of the fibre), but these cause problems for a straightforward application of the complex-weighted transformation, as the memory requirements scale $O(N_i^2 N_o^2)$.

### 5.1.3 Comparison of 1m and 10m results

While transmission of arbitrary images over 1m of bent fibre is already significantly longer than previous comparable work in the optics community, our results using a 10m fibre go an order of magnitude beyond that and approach the realm of communications. Figure 9 shows stills from a video from [Muybridge, 1955] at 1m and 10m, highlighting generalisation to content quite different from the MNIST and fashion training data.

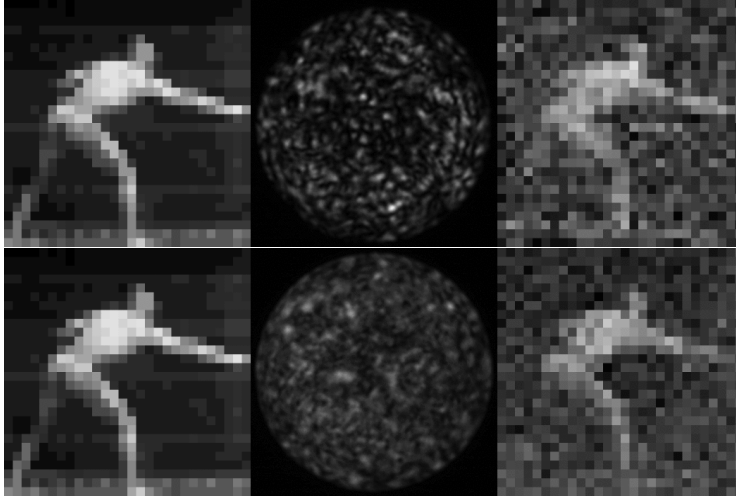

Figure 9: Comparison of inverse performance for 1m (upper) and 10m (lower) fibres, showing input, speckle image, inferred output. Note changes in speckle patterns for the longer distance—an increase in number, and decrease in size of speckles. Inverted images become noisier with increasing distance.

The outcome of the inverse transformation has inevitable errors—distributed speckled noise and systematic errors due to the missing phase information and the low precision of the SLM. Further improvements in specific domains which could be characterised in an appropriate training collection, could be gained by subsequent layers including, for example, the use of denoising convolutional autoencoder [Vincent et al., 2008]. Such denoising will become more valuable for longer distance fibres, where the losses in quality of the inversion from the complex layer become more significant.

## 6 Conclusions

We have presented an application of deep, complex-weighted networks to create the best known results on inversion of optical fibre image with a non-scanning, interferometer-free approach to sensing. This allows direct analogue image transfer over bent optical fibre at distances not achieved before without measurement of phase information. One concrete advantage of this approach is that it allows real-time video rate analogue image transmission along very thin ($105\,\mu m$) multi-mode fibre (scanning approaches would take $N^2$ longer to communicate each $N \times N$ image).

We contribute the *Optical fibre inverse problem*[6] benchmark dataset. This can act as an initial challenge set for machine learning researchers looking for an interesting challenge which can not be directly attacked by conventional convolutional networks. It brings challenging requirements of non-local patches on input and the need for better models of non-local relationships between pixels if we are to be able to work with the smaller speckles associated with longer optical fibres.

Despite these challenges, we achieved world-leading performance via the use of relatively simple, complex-weighted networks, which proved better than real-weighted networks in representing the inverse transmission matrix, and which can generalise to a wide range of images. Furthermore, we tested unitary, complex-weight regularisation, which improved performance compared to real-valued dense layers, is compatible with our physical understanding of the optical fibre inversion problem, and enables analytic invertibility of the trained network.

## Footnotes

[1]Note the non-locality of this system—the ability to project a focused spot in one specific position of the output relies on the knowledge of the whole matrix $T$, and would require inputs from all around the input image. This will create challenges when we apply memory-intensive deep networks to this problem. See Figure 1.

[2]After this NeurIPS submission, [Borhani et al., 2018, Fan et al., 2018] published deep, convolutional learning encoders inferring images from the speckle patterns, and demonstrate handwritten digit reconstruction. As pointed out in [Borhani et al., 2018], these approaches are more limited to images from the training classes. While Rahmani et al. [2018] do show simple binary images from outside the training set as examples of 'transfer learning', there is still limited evidence of true general, detailed imaging outside the training set.

[3]The approach used is monochrome, but RGB images can be communicated as monochrome channels through the complex transformation, and first integrated as RGB channels in later layers.

[4]Not to be confused with the Concatenated ReLU introduced in [Shang et al., 2016]

[5]We used the DSSIM implementation in the keras-contrib package. `https://github.com/keras-team/keras-contrib`

[6]Code and data can be found at `https://github.com/rodms/opticalfibreml`

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

**Acknowledgements**

We acknowledge funding from the Engineering and Physical Sciences Research Council on the *QuantIC* EPSRC grant EP/M01326X/1. R. Murray-Smith also acknowledges the support of the EPSRC *Closed-loop data science* grant EP/R018634/1. We would like to thank Francesco Tonolini and John Williamson for useful discussions and Maya Levitsky for discussions and support on simulation experiments.

