[Reviews · NeurIPS 2018]

Reviewer 1



The authors proposed a new unitary regularization approach for complex-weighted networks and demonstrate that it performs best in robustly inverting the fibres transmission matrix. In general, the problem statement is clear and appropriate for deep learning application. - The authors show the model comparison with different regularization. However, it would be good to compare with the other existing models, i.e., machine learning model, deep learning (like GANs) or physics-based model before they claim world-leading performance. - The proposed methods use two models, complex transformation, and autoencoder convnet. However, it is not clear to train autoencoder here. Denoising autoencoder is typically using raw images with adding noise and reconstruct the raw images. However, it is not clear how to train two models (i.e., complex transformation and autoencoder convnet) - Although the authors claim the proposed models fit well with the physical theory but it is not fully described. Further detailed information is required, for instance, complex-valued analogues, the relation between real and imaginary part in terms of training accuracy, etc.

Reviewer 2



I have read this paper and found it to be a solid contribution both to the field of optics and to NIPS. With respect to its contribution to optics, I will admit that I have not worked in this field for over two decades, but it does seem that they are solving an interesting problem in a new way. The removal of speckle from images seems to be a relevant problem in the field: (2018) SAR Image Despeckling Using a Convolutional Neural Network Puyang Wang, Student Member, IEEE, He Zhang, Student Member, IEEE and Vishal M. Patel, Senior Member, IEEE https://arxiv.org/pdf/1706.00552.pdf Most cited (2010) "General Bayesian estimation for speckle noise reduction in optical coherence tomography retinal imagery" Alexander Wong, Akshaya Mishra, Kostadinka Bizheva, David A. Clausi1 And I could not find a similar work in my search. The authors contribute compelling examples in their supplemental materials, which lends credence to the claims that their method actually works. The authors also promise to contribute a novel dataset to use for ML benchmarking on this type of problem. In my opinion, this paper comprises what I believe a NIPS application paper should have, a paper that would be considered best in class in it's field of application and one that uses neural information processing methods to bring a substantial advance to the application domain. Additionally the contribution of a benchmark dataset should further motivate research on this problem area. The paper is well written and easy to follow.

Reviewer 3



The authors present a complex valued deep convolutional network to invert the distortion effects induced by optical fibres when transmitting images. A potentially interesting dataset of images distorted by optical fibre transmission is presented for future benchmarking of algorithms trained on this task. An orthogonality regularizer for the complex convnet is presented that is shown to outperform simple l2 weight regularization. While I am not familiar with the field of application (optic fibre transmission of images), I think this paper presents a well presented and executed solution, demonstrating significant benefits of complex parameterization. My only concerns with this work are the lack of ablations. Since the real valued network appears to fail catastrophically, a small search over architectures with different forms of regularization besides L2 would make the point about complex parameterization stronger. Also, it was unclear to me from reading the introduction and related work if solutions to this problem have been presented by signal processing community in the past and how the methods presented compare to them.